# Cerebral hemodynamics in stroke thrombolysis (CHiST) study

**Man Y. Lam**[1]*, **Victoria J. Haunton**[1,2], **Ronney B. Panerai**[1,2], **Thompson G. Robinson**[1,2]

**1** Department of Cardiovascular Sciences, University of Leicester, Leicester, United Kingdom, **2** National Institute for Health Research Leicester Biomedical Research Centre, University of Leicester, Leicester, United Kingdom

* ml376@le.ac.uk

## Abstract

Despite careful patient selection, successful recanalization in intravenous thrombolysis is only achieved in approximately 50% of cases. Understanding changes in cerebral autoregulation during and following successful recanalization in acute ischemic stroke patients who receive intravenous thrombolysis, may inform the management of common physiological perturbations, including blood pressure, in turn reducing the risk of reperfusion injury. Cerebral blood velocity (Transcranial Doppler), blood pressure (Finometer) and end-tidal carbon dioxide (capnography) were continuously recorded in 11 acute ischemic stroke patients who received intravenous thrombolysis (5 female, mean ± SD age 68±12 years) over 4-time points, during and at the following time intervals after intravenous thrombolysis: 23.9±2.6 hrs, 18.1±7.0 days and 89.6±4.2 days. Reductions in blood pressure (p = 0.04) were observed during intravenous thrombolysis. Reductions in heart rate (p<0.005) and critical closing pressure [Affected hemisphere (p = 0.02) and non-affected hemisphere (p<0.005)] were observed post intravenous thrombolysis. End-tidal $CO_2$ increased during the subacute and chronic stages (p = 0.028). Reduction in affected hemisphere phase at low frequency was observed during intravenous thrombolysis (p = 0.021) and at subsequent visits (p = 0.048). No changes were observed in cerebral blood velocity, coherence, gain and Autoregulation Index during the follow-up period. Intravenous thrombolysis in acute ischemic stroke patients induced changes in affected hemisphere phase and other key hemodynamic parameters, but not Autoregulation Index. Further investigation of cerebral autoregulation is warranted in a larger acute ischemic stroke cohort to inform its potential role in individualized management plans.

## Introduction

One of the fundamental aims in hyperacute stroke management is to rapidly restore cerebral blood flow (CBF) to ischemic, but potentially, viable tissue; the so-called penumbra. This is achieved by reperfusion of completely or partially occluded intracranial arteries or by improving collateral flow, either using pharmacological (e.g. pressor therapy, IVT) or non-pharmacological means (e.g. mechanical thrombectomy (MT), head positioning adjustment). IVT is

**Data Availability Statement:** All relevant data are within the manuscript and its supporting information files.

**Funding:** The author(s) received no specific funding for this work.

**Competing interests:** TGR is an NIHR Senior Investigator, this does not alter our adherence to PLOS ONE policies on sharing data and materials.

approved for medical reperfusion therapy in AIS, within 4.5 hours of stroke symptom onset [1, 2]. Despite significant recent increases in thrombolysis rates [3, 4], improvement in selecting eligible participants, and availability of expertise and associated technologies, successful recanalization of occluded intracranial arteries with IVT is achieved in only approximately 50% of AIS patients [5]. Studies have suggested that without careful patient selection, angiographic recanalization may not be associated with meaningful improvements in clinical outcomes [6, 7]. Importantly, thrombolysis is associated with risks of reperfusion injury, such as post-ischemic edema and/or symptomatic intracerebral hemorrhage (sICH).

Age, degree of initial neurological deficit, timing of reperfusion therapy, size of baseline infarct, location of occlusion, and presence of persistent or re-occlusion and hemorrhagic transformation [8–10] are some of the factors which could predict the chance of success in recanalization, and therefore, stroke outcome. Limited evidence is available on systemic arterial BP, an important factor which has significant impact on the penumbral lesion size, and its association with neurological recovery. Moreover, little information is available on whether optimal personalised BP could be one of the strategies to reduce complications of reperfusion, and therefore improve clinical outcome. Systolic BP has been demonstrated as an important predictor of sICH [11] and the recently published ENCHANTED trial reported that intensive BP lowering was associated with reduced rates of any intracranial hemorrhage but without benefit on 90-day functional outcomes [12]. Impairment in CA, in association with increases in BP variability (BPV), means that large fluctuations in BP may lead to changes in CBF outside the CA plateau capacity and result in hyperemic or ischemic injury [13, 14]. There is limited knowledge regarding the natural history and prognostic significance of impaired CA during and following recanalization and reperfusion, especially in those patients who received IVT treatment, and few studies have investigated the temporal relationship of CA during and immediately after IVT treatment in AIS patients. Therefore, the aim of this study was to look at how CA, and associated systemic and cerebral hemodynamic parameters, in AIS patients respond to IVT. In particular, we looked at these changes during and immediately after IVT, and for up to 3 months of AIS symptom onset, so that we could investigate whether such changes were associated with any improvement or deterioration in neurological and functional outcome, so providing evidence to inform individualized BP management decisions in the future.

## Materials and methods

### Research participants

The study was carried out in accordance with the Declaration of Helsinki (2000) and was approved by the Nottingham Research Ethics Committee 1 (Reference: 15/EM/0485). Clinical Trial.gov unique identifier: NCT 02928926.

Eleven AIS patients were recruited from the University Hospitals of Leicester (UHL) NHS Trust, Leicester, United Kingdom. AIS patients were transferred by Rapid Ambulance Protocol to a dedicated hyperacute stroke unit (HASU). Patients who met the criteria for thrombolytic therapy with IVT [15] were invited to participate in the study. AIS patients who were deemed not eligible for IVT, had premorbid modified Rankin Scale (mRS) [16] >3, or co-morbidity with anticipated life expectancy <3 months, or currently participating in another investigational drug trial, were excluded from the study.

Clinical stroke subtype, stroke severity, and pre-morbid functional dependence were assessed using the Oxfordshire Community Stroke Project (OCSP) classification [17], National Institutes of Health Stroke Scale (NIHSS) [18], and mRS, respectively. Participants' handedness was determined by the Edinburgh Inventory [19]. All routine aspects of the management

of AIS patients with respect to investigation, hyperacute and acute management, and rehabilitation were continued according to both national and local hospital guidelines. These included IVT at a dose of 0.9 mg/kg/body weight (upper limit 90 mg) as a continuous infusion over 60 mins, with 10% of the total dose administered as a bolus. Non-contrast Computed Tomography (CT) imaging was carried out pre and 24 hours post IVT treatment. BP measurements according to the standard thrombolysis protocol, were recorded throughout the immediate 24-hour period following the initiation of IVT therapy, i.e. every 15 minutes for 2 hours, then every 30 minutes for 6 hours, and subsequently hourly for 16 hours. All AIS patients received pharmacological treatment, including antithrombotic, anticoagulant, antihypertensive and statin therapy according to hospital protocols.

All participants or personal consultees provided informed consent in writing, and were aware of the right to withdraw from the study at any point in time without prejudice.

## Procedure

Efforts were made to ensure all AIS assessments were carried out at a similar time of day at all visits. The study was carried out either in the HASU (Visit 1.1–1.4) or dedicated research laboratory (Visit 2 onwards), which was at a controlled temperature (20–24˚C) with minimal external distraction. All parameters were recorded with participants in the lying flat (0˚) head position. The BP signal was calibrated at the beginning of each recording using the brachial BP readings taken by an OMRON 705IT sphygmomanometer. Continuous BP was measured non-invasively using finger arterial volume clamping (Finometer®, Finapres Medical Systems; Amsterdam, Netherlands) attached to the middle finger of the non-hemiparetic hand. The servo-correcting mechanism of the Finometer® was switched on then off prior to measurements. The cuff was held at heart level to minimize any orthostatic pressure difference between finger and heart. A 3-lead electrocardiogram (ECG) was used to record heart rate (HR). Respiratory rate and ETCO$_2$ were monitored using an infrared capnograph (Capnocheck Plus, Smith Medical, Minnesota, USA) attached to a nasal cannula (Salter Labs, California, USA). The middle cerebral arteries (MCAs) were insonated bilaterally using transcranial Doppler (TCD) ultrasonography (Viasys Companion III, Natus Medical Incorporated, California, USA). Two transducers operating at 2 MHz were positioned on the temporal bone and the proximal segment of the MCA was identified according to the depth, waveform and characteristics of the signal [20]. The depth, power, velocity, and location of the temporal window were documented in each participant to ensure the same measuring parameters were used at subsequent visits. A custom-made head frame was used to secure the ultrasound probes in position and to minimize movement. All TCD measurements were carried out by the first author who has been formally trained with significant experiences in operating TCD and published reproducibility data [21].

Once the MCAs had been identified and the ultrasound probes were secured in place, four 5-min measurements were recorded during (Visit 1.1 and 1.2), immediately prior to completion (Visit 1.3), and immediately after the completion (Visit 1.4) of IVT. Subsequent assessments were undertaken at approximately 24 hours (Visit 2, acute), 2 weeks (Visit 3, sub-acute) and 3 months (Visit 4, chronic) post stroke symptom onset.

## Data analysis

All measurements were simultaneously recorded onto a physiological data acquisition system (PHYSIDAS, Department of Medical Physics, UHL, Leicester, United Kingdom) at a rate of 500 samples/s, for subsequent off-line editing and analysis. All signals were visually inspected to identify any artefacts and noise, and BP waveforms were also inspected for any drift; narrow

spikes (<100ms) were removed by linear interpolation. Recordings were rejected if there was a BP drift, excessive artefact, noise, or poor quality of cerebral blood velocity (CBV) signals. The CBV signals were subjected to a median filter. All signals were then low pass filtered with a zero-phase, Butterworth filter with cut-off frequency of 20Hz. The ECG was marked to determine the R-R interval, and continuous HR was plotted against time. Any ectopic beats, resulting in spikes in the HR signal, were identified by visual inspection of the affected QRS complex. These were then manually corrected by remarking R-R intervals for the time points at which they occurred. In cases where there was ECG drift or noise, cardiac cycles were marked using the BP tracing as an alternative. Mean, systolic, and diastolic BP, CBV and HR were calculated for each cardiac cycle. Critical closing pressure (CrCP) and resistance area product (RAP) were estimated using the first harmonic method [22]; these parameters being calculated to reflect the instantaneous CBV-BP relationship changes during and after IVT infusion. Linear interpolation was used to obtain estimates of $ETCO_2$ synchronized to the end of each cardiac cycle. Beat-to-beat data were spline interpolated, with resampling at 5Hz to produce signals with a uniform time base.

## Transfer function analysis

Transfer function analysis (TFA) was adopted with parameter settings recommended by the International Cerebral Autoregulation Research Network (CARNet) [23] to estimate the CBV response to a hypothetical step change in BP. Each of the 10 template CBV step responses proposed by Tiecks et al. [24] was compared with the estimated CBV step response and the autoregulation index (ARI) value corresponding to the best fit, assessed by the minimum mean square error, was adopted for each of the 5-min recordings.

Estimates of coherence, phase, and gain were averaged for the very low frequency (VLF; 0.02–0.07 Hz), low frequency (LF; 0.07–0.20 Hz), and high frequency (HF; 0.20–0.50 Hz) ranges [23]. In brief, phase and gain represent the temporal and amplitude relationship between input (BP) and output (CBV) at each frequency, whereas coherence represents the linearity between input and output; coherence approaching 1.0 indicate a linear relationship; whereas coherence approaching zero could indicate a non-linear relationship, poor signal-to-noise ratio or other inputs also influencing CBV changes. In general, a lower gain and higher phase indicate a more effective CA response [25].

## Statistical analysis

Data were assessed for normality using the Shapiro-Wilk test. All normality distributed data are presented as mean ± SD, and continuous skewed data as median [interquartile range]. Mean values of hemodynamic parameters were calculated from each of the 5-min recordings. Two-way repeated measures ANOVA was undertaken to assess the effects of time (Intra-visit: Visit 1.1 to 1.4 and Inter-visit: Visit 1.4 to 4) and affected (AH) vs. non-affected hemisphere (NAH). The Tukey post hoc test was used to perform individual comparisons when ANOVA showed significant effects. A significance level of $p < 0.05$ was adopted for all results.

A formal sample size calculation was not possible for the study. However, to detect a change of 2 units in ARI and to estimate the required sample size we followed the results previously described by Brodie and colleagues [26]. For this study, a sample of 11 AIS patients will allow the detection of a difference in the ARI of 2 units with 80% power at the 5% significance level. All statistical analyses were performed using TIBCO Statistica, version 13.0 (Statistica, Dell).

## Results

Fifteen participants (7 female) were recruited, though suitable transtemporal windows could not be found in four participants (2 female) who were removed from the study. Therefore, a total of 11 AIS patients of mean age 68 years (range 42 to 80) were included in the analysis and completed the hyperacute, acute, sub-acute, and chronic visits at $132 \pm 38$ mins, $23.9 \pm 2.6$ hrs, $18.1 \pm 7.0$ days, and $89.6 \pm 4.2$ days after stroke onset, respectively. According to the OCSP classification, there were three total and six partial anterior circulation, one posterior circulation, and one lacunar stroke. According to the Edinburgh Handedness inventory [19], 10 participants were right-handed and one participant was left-handed. Mean baseline NIHSS and median pre-admission mRS scores at the first visit were $8.2 \pm 3.8$ and 0 [0–1], respectively. No patients received MT and none of the patients received antihypertensive or analgesia during IVT. One patient required supplemental oxygen during IVT (one litre of oxygen via nasal cannula). Four patients discharged with additional antihypertensive medication, with two additional patients receiving antihypertensive medication at Visit 4 (3 months post stroke symptom onset).

Participant characteristics are further summarized in Table 1.

### Temporal pattern of systemic hemodynamic and $CO_2$ responses

Systemic hemodynamic and other parameters averaged over 5-min are given in Table 2 for each visit. Mean arterial BP (MAP) decreased over four visits (Visit 1.1: $110.7 \pm 16.3$ to Visit 4: $94.0 \pm 9.4$ mmHg), although significant reduction was only observed during IVT ($p = 0.044$, Table 2 and Fig 1A). Significant reduction in HR was observed post IVT (Visit 1.4: $81.2 \pm 15.7$ to Visit 4: $62.8 \pm 14.9$ bpm; $p<0.005$; Table 2 and Fig 1B). Hypocapnia was observed during the hyperacute phase of AIS and $ETCO_2$ did not show significant changes during or immediately after IVT infusion, or within the first 24 hours of stroke onset. However, significant increments were observed in subsequent visits (Visit 3: $35.5 \pm 2.5$ and Visit 4: $36.5 \pm 1.7$ mmHg; $p = 0.028$; Table 2 and Fig 1C).

### Temporal pattern of cerebral hemodynamic responses

Cerebral hemodynamic parameters averaged over 5-min are also given in Table 2 for each visit. CBV and RAP did not change across four visits, and there were no differences between hemispheres either (Table 2 and Fig 1D and 1F, respectively). However, a non-significant rise in CrCP was demonstrated in both the AH and NAH, immediately after the completion of IVT, followed by a significant reduction over subsequent visit (AH Visit 1.4: $55.2 \pm 12.9$ to visit 4: $43.4 \pm 8.3$ mmHg; $p = 0.02$ and NAH Visit 1.4: $60.1 \pm 12.6$ to Visit 4: $40.9 \pm 8.6$ mmHg; $p<0.005$, Table 2 and Fig 1E). No differences between hemispheric values were observed across all four visits.

Though NAH CBV was numerically higher, compared to AH CBV across all four visits, differences did not reach statistical significance (Table 2 and Fig 1D). There were no significant changes with respect to AH CBV values during and immediately after IVT or across subsequent visits (Table 2 and Fig 1D). There were also no changes with respect to NAH CBV values during and immediately after IVT or across subsequent visits (Table 2 and Fig 1D).

Similar to CBV, there were no differences between AH and NAH RAP values across all four visits, nor any differences between hemispheres (Table 2 and Fig 1F).

### Temporal pattern of coherence, gain, and phase responses

Averaged 5-min assessment values of coherence, gain, and phase at different frequency intervals (VLF, LF, and HF) are given in Tables 3–5, respectively. There were no differences in

**Table 1. Demographic characteristics of participants.**

| | |
|---|---|
| **Participants (n)** | **11** |
| **Age (years)** | **68.0 ± 11.6** |
| **Sex (female), n (%)** | **5 (45.4)** |
| **Handedness (right), n (%)** | **10 (90.9)** |
| **Body mass index (BMI) kg.m$^{-2}$** | **27.1 ± 4.5** |
| **Smoker, n (%)** | |
| Yes | 2 (18.2) |
| Ex | 4 (36.4) |
| Never | 5 (45.5) |
| **Past medical history, n (%)** | |
| Hypertension | 4 (36.4) |
| Hypercholesterolemia | 2 (18.2) |
| Diabetes | 2 (18.2) |
| Ischemic Heart Disease | 2 (18.2) |
| **NIHSS admission** | **8.2 ± 3.8** |
| Post thrombolysis | 5.2 ± 3.9 |
| Visit 2 | 3.0 ± 2.9 |
| Visit 3 | 0 [0–2] |
| Visit 4 | 1 [0–4] |
| **mRS pre admission** | **0 [0–1]** |
| Visit 2 | 1 [1–2] |
| Visit 3 | 1 [0–1] |
| Visit 4 | 2 [1–2] |
| **Time of assessment** | |
| Visit 1 | 132 ± 38 mins |
| Visit 2 | 23.9 ± 2.6 hrs |
| Visit 3 | 18.1 ± 7.0 days |
| Visit 4 | 89.6 ± 4.2 days |
| **OCSP Classification** | |
| TACS | 3 |
| PACS | 6 |
| LACS | 1 |
| POCS | 1 |
| **Handedness** | |
| Right | 10 |
| Left | 1 |

Data are mean ± SD, median [IQR], or n (%). NIHSS, National Institutes of Health Stroke Scale; mRS, modified Rankin score.

overall temporal pattern for coherence and gain across all four visits, in VLF (Coherence: Fig 2A and Gain: Fig 2D), LF (Coherence: Fig 2B and Gain: Fig 2E) and HF (Coherence: Fig 2C and Gain: Fig 2F), nor were there any inter-hemispheric differences (Tables 3–5; Figs 2 and 3). However, a reduction in phase (AH) at LF occurred during IVT (Visit 1.1: 0.89 ± 0.42 to Visit 1.4: 0.63 ± 0.26 radians; p = 0.021), such reduction continued post IVT, and was followed by an increment at 3 months post stroke symptom onset (Visit 4: 0.70 ± 0.31 radians; p = 0.048) (Table 5 and Fig 3B).

**Table 2. Systemic and cerebral hemodynamic parameters in AIS patients receiving IVT.**

| Parameters | Visit 1.1 n = 11 | Visit 1.2 n = 11 | Visit 1.3 n = 10 | Visit 1.4 n = 11 | Visit 2 n = 11 | Visit 3 n = 8 | Visit 4 n = 7 | P value (Intra-visit effect Visit 1.1–1.4) | P value (Inter-visit ffect Visit 1.4–4) | P value (Hemispheric effect, Visit 1.1–1.4) | P value (Hemispheric effect, Visit 1.4–4) |
|---|---|---|---|---|---|---|---|---|---|---|---|
| BP (mmHg) | 110.7 ± 16.3 | 103.7 ± 15.6 | 101.1 ± 17.1* | 102.6 ±16.3 | 99.1 ± 8.3 | 97.3 ± 7.0 | 94.0 ± 9.4 | 0.044 | 0.13 | | |
| Heart Rate (bpm) | 78.2 ± 14.9 | 78.8 ± 15.5 | 79.2 ± 15.5 | 81.2 ± 15.7 | 72.4 ± 15.5† | 66.6 ± 17.1† | 62.8 ± 14.9†‡ | 0.12 | <0.005 | | |
| End-Tidal $CO_2$ (mmHg) | 31.9 ± 3.9 | 32.7 ± 4.4 | 31.2 ± 4.9 | 31.4 ± 4.9 | 32.6 ± 4.3 | 35.5 ± 2.5 | 36.5 ± 1.7† | 0.39 | 0.028 | | |
| CBV AH (cm.s$^{-1}$) | 35.5 ± 10.9 | 35.9 ± 9.7 | 36.5 ± 9.1 | 35.3 ± 7.1 | 37.9 ± 7.0 | 35.6 ± 8.9 | 39.9 ± 9.4 | 0.99 | 0.29 | 0.58 | 0.089 |
| CBV NAH (cm.s$^{-1}$) | 41.1 ± 13.1 | 38.6 ± 11.6 | 37.9 ± 11.4 | 38.7 ± 11.6 | 41.2 ± 13.2 | 42.1 ± 7.4 | 43.2 ± 8.4 | 0.15 | 0.64 | | |
| CrCP AH (mmHg) | 53.0 ± 18.6 | 52.9 ± 11.4 | 52.3 ± 12.8 | 55.2 ± 12.9 | 48.0 ± 10.9 | 45.1 ± 10.9 | 43.3 ± 8.3† | 0.69 | 0.02 | 0.24 | 0.08 |
| CrCP NAH (mmHg) | 54.8 ± 15.1 | 49.1 ± 8.8 | 51.5 ± 4.5 | 60.1 ± 12.6 | 48.2 ± 10.0† | 45.0 ± 6.9† | 40.9 ± 8.6† | 0.12 | <0.005 | | |
| RAP AH (mmHg.s.cm$^{-1}$) | 1.50 ± 0.71 | 1.37 ± 0.64 | 1.30 ± 0.53 | 1.32 ± 0.46 | 1.25 ± 0.43 | 1.55 ± 0.58 | 1.35 ± 0.61 | 0.27 | 0.38 | 0.26 | 0.29 |
| RAP NAH (mmHg.s.cm$^{-1}$) | 1.56 ± 0.75 | 1.34 ± 0.47 | 1.35 ± 0.44 | 1.31 ± 0.41 | 1.30 ± 0.50 | 1.29 ± 0.29 | 1.20 ± 0.31 | 0.40 | 0.92 | | |

Data are mean ± SD. P-values from two-way ANOVA for effects of during and immediately post-IVT (Intra-visit: Visit 1.1 to Visit 1.4) and subsequent visits (Inter-visit: Visit 1.4 to Visit 4) (Visit effect) and for the effects of hemisphere (AH vs. NAH). AIS, acute ischemic stroke; BP, blood pressure; bpm, beats per minutes; CBV, cerebral blood velocity; CrCP, critical closing pressure; RAP, resistance area product; AH, affected hemisphere; IVT, intravenous thrombolysis NAH, non-affected hemisphere. Visit 1.1 and Visit 1.2 refer to during IVT, Visit 1.3 refers to immediately prior to end of IVT, and Visit 1.4 refers to immediately after the end of IVT. Visit 2, Visit 3 and Visit 4 refer to approximately 24 hours, 2 weeks, and 3 months post stroke symptom onset, respectively.

* Tukey post hoc compared to Visit 1.1,

† Tukey post hoc compared to Visit 1.4 and

‡ Tukey post hoc compared to Visit 2, all p <0.05.

## Temporal pattern of ARI responses

Overall, there were no changes in ARI values during IVT, across all visits, nor between AH and NAH (Table 6 and Fig 3D).

Fig 4 shows individual AIS patients' ARI at each visit. There was no obvious trend demonstrated in terms of ARI values between AH (Fig 4A) and NAH (Fig 4B) across all visits.

## Discussion

### Main findings

This longitudinal study assessed serial TCD and other physiological measurements in CA, with associated systemic and cerebral hemodynamic parameters, during and after IVT administration, up to 3 months of post stroke symptom onset. Previous reports suggested CA changes in AIS patients tend to occur within days and weeks post stroke symptom onset [27, 28], we chose to carry out 3 months follow up to correlate any CA changes with neurological recovery and functional outcome.

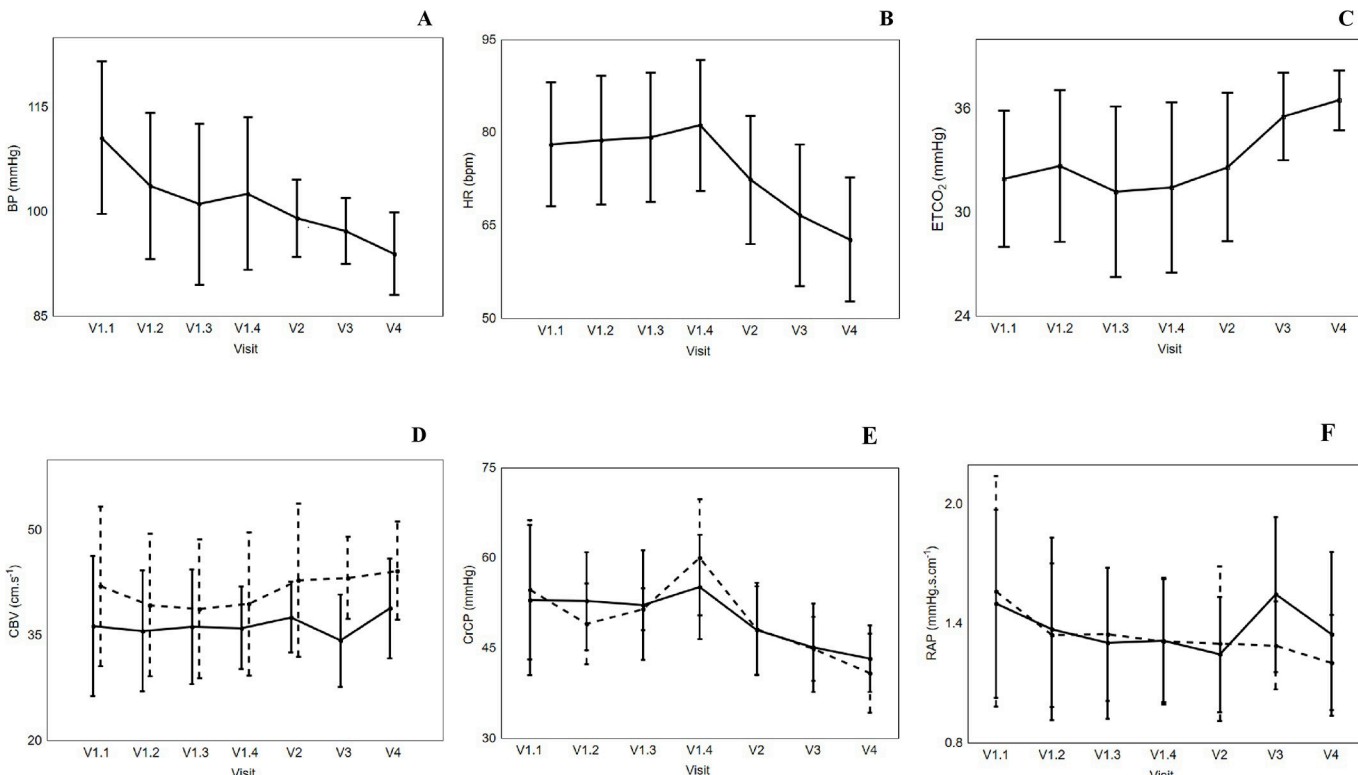

**Fig 1.** Effects of visit on the changes in blood pressure (BP) [A], heart rate (HR) [B] and end-tidal $CO_2$ [C]. Vertical bar denotes 95% confidence interval. V1.1 and V1.2 refer to during IVT administration, V1.3 refers to immediately prior to end of IVT, and V1.4 refers to immediately after the end of IVT. V2, V3 and V4 refers to approximately 24 hours, 2 weeks and 3 months post stroke symptom onset, respectively. Effects of visit on [D] changes in cerebral blood velocity (CBV), [E] critical closing pressure (CrCP) and [F] resistance area product (RAP). Affected hemisphere (continuous line) and non-affected hemisphere (dotted line). Vertical bar denotes 95% confidence interval. V1.1 and V1.2 refer to during IVT administration, V1.3 refers to immediately prior to end of IVT, and V1.4 refers to immediately after the end of IVT. V2, V3 and V4 refers to approximately 24 hours, 2 weeks and 3 months post stroke symptom onset, respectively.

The study demonstrated a gradual reduction in BP and HR across visits, which is in broad agreement with other literature [29–32]. It is important to note that significant BP reduction occurred during IVT, rather than subsequent visits, highlighting the importance of having BP closely monitored during this critical period. When compared to healthy controls [33], profound hypocapnia was observed during and immediately after completion of IVT, and up to 24 hours after stroke onset, with significant increment in subsequent visits. These later increments in $ETCO_2$ coincided with a reduction in BP, HR and CrCP. However, vasodilatation, and increases in CBF and sympathetic activity, BP and HR are more usually associated with normalisation of hypocapnia [33, 34]. Therefore, the reduction observed in BP, could be due to other reasons such as recovery from pain and discomfort secondary to AIS event, patient 'reassurance' through management in a specialist HASU environment, changes in neuro-endocrine hormones [35], transient autonomic nervous system imbalance [36], baroreceptor sensitivity [37], or Cushing's reflex [38].

## Cerebral hemodynamic changes following IVT

In general, there were no significant differences in both AH and NAH CBV changes during and immediately after IVT, and at subsequent follow-up visits. Conflicting results have been demonstrated in the literature; for example, Reinhard et al. undertook a TCD study of the MCA in AIS patients, with a significant proportion of patients receiving IVT therapy (42%).

**Table 3. Coherence in various frequencies (VLF, LF, HF) in AIS receiving IVT.**

| Parameters | Visit 1.1 n = 11 | Visit 1.2 n = 11 | Visit 1.3 n = 10 | Visit 1.4 n = 11 | Visit 2 n = 11 | Visit 3 n = 8 | Visit 4 n = 7 | P value (Intra-visit effect, Visit 1.1–1.4) | P value (Inter-visit effect, Visit 1.4–4) | P value (Hemispheric value, Visit 1.1–1.4) | P value (Hemispheric value, Visit 1.4–4) |
|---|---|---|---|---|---|---|---|---|---|---|---|
| Coherence VLF, AH | 0.46 ± 0.13 | 0.33 ± 0.21 | 0.48 ± 0.14 | 0.37 ± 0.16 | 0.51 ± 0.21 | 0.51 ± 0.21 | 0.61 ± 0.12 | 0.18 | 0.065 | 0.17 | 0.47 |
| Coherence VLF, NAH | 0.45 ± 0.16 | 0.42 ± 0.17 | 0.36 ± 0.17 | 0.38 ± 0.16 | 0.50 ± 0.15 | 0.43 ± 0.21 | 0.60 ± 0.13 | 0.41 | 0.057 | | |
| Coherence LF, AH | 0.47 ± 0.20 | 0.40 ± 0.21 | 0.45 ± 0.25 | 0.46 ± 0.17 | 0.50 ± 0.23 | 0.54 ± 0.26 | 0.63 ± 0.27 | 0.80 | 0.057 | 0.82 | 0.17 |
| Coherence LF, NAH | 0.58 ± 0.20 | 0.51 ± 0.28 | 0.57 ± 0.19 | 0.56 ± 0.19 | 0.56 ± 0.22 | 0.67 ± 0.20 | 0.69 ± 0.27 | 0.93 | 0.15 | | |
| Coherence HF, AH | 0.52 ± 0.19 | 0.49 ± 0.26 | 0.56 ± 0.28 | 0.52 ± 0.15 | 0.63 ± 0.22 | 0.60 ± 0.25 | 0.66 ± 0.27 | 0.65 | 0.43 | 0.91 | 0.23 |
| Coherence HF, NAH | 0.57 ± 0.27 | 0.56 ± 0.25 | 0.61 ± 0.26 | 0.60 ± 0.24 | 0.71 ± 0.18 | 0.70 ± 0.22 | 0.75 ± 0.27 | 0.90 | 0.40 | | |

Data are mean ± SD. P-values from two-way ANOVA for effects of during and immediately post-IVT (Intra-visit: Visit 1.1 to Visit 1.4) and subsequent visits (Inter-visit: Visit 1.4 to Visit 4) (Visit effect) and for the effects of hemisphere (AH vs. NAH). AH, affected hemisphere; AIS, acute ischemic stroke; HF, high frequency; IVT, intravenous thrombolysis; LF, low frequency; NAH, non-affected hemisphere; VLF, very low frequency. Visit 1.1 and Visit 1.2 refer to during IVT, Visit 1.3 refers to immediately prior to end of IVT, and Visit 1.4 refers to immediately after the end of IVT. Visit 2, Visit 3, and Visit 4 refer to approximately 24 hours, 2 weeks, and 3 months post stroke symptom onset, respectively.

**Table 4. Gain in various frequencies (VLF, LF, HF) in AIS patients receiving IVT.**

| Parameters | Visit 1.1 n = 11 | Visit 1.2 n = 11 | Visit 1.3 n = 10 | Visit 1.4 n = 11 | Visit 2 n = 11 | Visit 3 n = 8 | Visit 4 n = 7 | P value (Intra-visit effect, Visit 1.1–1.4) | P value (Inter-visit effect, Visit 1.4–4) | P value (Hemispheric effect, Visit 1.1–1.4) | P value (Hemispheric effect, Visit 1.4–4) |
|---|---|---|---|---|---|---|---|---|---|---|---|
| Gain VLF, AH (cm s$^{-1}$ mmHg$^{-1}$) | 0.40 ± 0.22 | 0.53 ± 0.36 | 0.52 ± 0.31 | 0.51 ± 0.35 | 0.65 ± 0.44 | 0.76 ± 0.52 | 0.68 ± 0.46 | 0.34 | 0.11 | 0.29 | 0.29 |
| Gain VLF, NAH (cm s$^{-1}$ mmHg$^{-1}$) | 0.56 ± 0.23 | 0.51 ± 0.30 | 0.55 ± 0.33 | 0.79 ± 0.39 | 0.53 ± 0.20 | 0.64 ± 0.34 | 0.66 ± 0.38 | 0.73 | 0.41 | | |
| **Gain LF, AH (cm s$^{-1}$ mmHg$^{-1}$)** | 0.53 ± 0.33 | 0.67 ± 0.62 | 0.78 ± 0.67 | 0.73 ± 0.52 | 0.82 ± 0.45 | 0.81 ± 0.52 | 0.92 ± 0.56 | 0.28 | 0.24 | 0.49 | 0.13 |
| Gain LF, NAH (cm s$^{-1}$ mmHg$^{-1}$) | 0.73 ± 0.42 | 0.66 ± 0.32 | 0.77 ± 0.32 | 0.81 ± 0.38 | 0.76 ± 0.26 | 0.73 ± 0.22 | 0.82 ± 0.22 | 0.54 | 0.89 | | |
| Gain HF, AH (cm s$^{-1}$ mmHg$^{-1}$) | 0.66 ± 0.35 | 0.90 ± 0.73 | 0.94 ± 0.47 | 1.04 ± 0.82 | 0.99 ± 0.37 | 0.85 ± 0.37 | 0.99 ± 0.49 | 0.13 | 0.54 | 0.46 | 0.70 |
| Gain HF, NAH (cm s$^{-1}$ mmHg$^{-1}$) | 0.82 ± 0.54 | 0.83 ± 0.50 | 1.00 ± 0.56 | 0.85 ± 0.26 | 0.85 ± 0.45 | 1.04 ± 0.31 | 1.07 ± 0.43 | 0.69 | 0.40 | | |

Data are mean ± SD. P-values from two-way ANOVA for effects of during and immediately post-IVT (Intra-visit: Visit 1.1 to Visit 1.4) and subsequent visit (Inter-visit: Visit 1.4 to Visit 4) (Visit effect) and for the effects of hemisphere (AH vs. NAH). AH, affected hemisphere; AIS, acute ischemic stroke; HF, high frequency; IVT, intravenous thrombolysis; LF, low frequency; NAH, non-affected hemisphere; VLF, very low frequency. Visit 1.1 and Visit 1.2 refer to during IVT, Visit 1.3 refers to immediately prior to end of IVT, and Visit 1.4 refers to immediately after the end of IVT. Visit 2, Visit 3, and Visit 4 refer to approximately 24 hours, 2 weeks, and 3 months post stroke symptom onset, respectively.

**Table 5. Phase in various frequencies (VLF, LF, HF) in AIS patients receiving IVT.**

| Parameters | Visit 1.1 | Visit 1.2 | Visit 1.3 | Visit 1.4 | Visit 2 | Visit 3 | Visit 4 | P value | P value | P value | P value |
|---|---|---|---|---|---|---|---|---|---|---|---|
| | n = 11 | n = 11 | n = 10 | n = 11 | n = 11 | n = 8 | n = 7 | (Intra-visit effect, Visit 1.1–1.4) | (Inter-visit effect, Visit 1.4–4) | (Hemispheric effect, Visit 1.1–1.4) | (Hemispheric effect, Visit 1.4–4) |
| Phase VLF, AH (Radians) | 0.69 ± 0.43 | 0.50 ± 0.29 | 0.49 ± 0.33 | 0.48 ± 0.41 | 0.59 ± 0.38 | 0.77 ± 0.40 | 0.79 ± 0.34 | 0.53 | 0.18 | 0.53 | 0.81 |
| Phase VLF, NAH (Radians) | 0.83 ± 0.34 | 0.70 ± 0.22 | 0.81 ± 0.27 | 0.71 ± 0.32 | 0.74 ± 0.42 | 0.83 ± 0.38 | 0.76 ± 0.24 | 0.70 | 0.85 | | |
| Phase LF, AH (Radians) | 0.89 ± 0.42 | 0.60 ± 0.42* | 0.65 ± 0.40 | 0.63 ± 0.26* | 0.54 ± 0.22 | 0.44 ± 0.28 | 0.70 ± 0.31§ | 0.021 | 0.048 | 0.24 | 0.87 |
| Phase LF, NAH (Radians) | 0.72 ± 0.21 | 0.68 ± 0.36 | 0.68 ± 0.27 | 0.72 ± 0.40 | 0.51 ± 0.11 | 0.46 ± 0.22 | 0.44 ± 0.21 | 0.97 | 0.065 | | |
| Phase HF, AH (Radians) | 0.047 ± 0.26 | -0.061 ± 0.23 | 0.051 ± 0.21 | 0.086 ± 0.18 | 0.045 ± 0.086 | 0.088 ± 0.14 | 0.047 ± 0.15 | 0.38 | 0.80 | 0.52 | 0.58 |
| Phase HF, NAH (Radians) | 0.012 ± 0.11 | -0.031 ± 0.20 | -0.074 ± 0.22 | 0.094 ± 0.13 | 0.069 ± 0.088 | 0.066 ± 0.062 | 0.11 ± 0.18 | 0.10 | 0.81 | | |

Data are mean ± SD. P-values from two-way ANOVA for effects of during and immediately post-IVT (Intra-visit: Visit 1.1 to Visit 1.4) and subsequent visit (Inter-visit: Visit 1.4 to Visit 4) (Visit effect) and for the effects of hemisphere (AH vs. NAH). AH, affected hemisphere; AIS, acute ischemic stroke; HF, high frequency; IVT, intravenous thrombolysis; LF, low frequency; NAH, non-affected hemisphere; VLF, very low frequency. Visit 1.1 and Visit 1.2 refer to during IVT, Visit 1.3 refers to immediately prior to end of IVT, and Visit 1.4 refers to immediately after the end of IVT. Visit 2, Visit 3, and Visit 4 refer to approximately 24 hours, 2 weeks, and 3 months post stroke symptom onset, respectively.

* Tukey post hoc compared to Visit 1.1,

§ Tukey post hoc compared to Visit 3, all p < 0.05.

This study reported a significant increment in both AH and NAH CBV, approximately 7 days post stroke symptom onset. Interestingly, it was associated with a reduction in phase, suggesting worsening of dCA within first week of onset [39]. However, Petersen and colleagues demonstrated a reduction in both AH and NAH CBV, at day 3–6 of stroke symptom onset, which slowly returned to baseline values at subsequent visits [31]. Of note, Akopov et al. found a wide range of serial CBV results in AIS patients, and suggested that this may be caused by the degree of intra-cranial arterial stenosis, occlusion and recanalization [40]. Interestingly, despite improvement in both stroke severity (NIHSS) and functional outcome (mRS) at 3 months in our AIS cohort, there were not any significant changes in terms of CBV between both hemispheres and over time. This is in agreement with the recent study carried out by Salinet et al. who assessed more than 50 ischemic stroke patients up to 31 hours post symptom onset. It was reported that patients with mild (median NIHSS = 2) and moderate (median NIHSS = 9) stroke severity, there was no significant differences between AH and NAH CBV [41]. AIS patients in the present study had an admission mean NIHSS of 8.2, and this may therefore explain the lack of significant differences between AH and NAH CBV. However, in the absence of angiographic data, it is difficult to determine whether there was no occlusion at presentation and no recanalization at follow-up, or alternatively complete MCA occlusion without successful recanalization. Future studies should therefore include angiography to assess the degree of occlusion, recanalization and response of treatment.

CrCP represents the value of arterial BP where cerebral blood vessels collapse and CBF ceases [22, 42], and is often used as an index of cerebral vascular tone [43]. RAP reflects

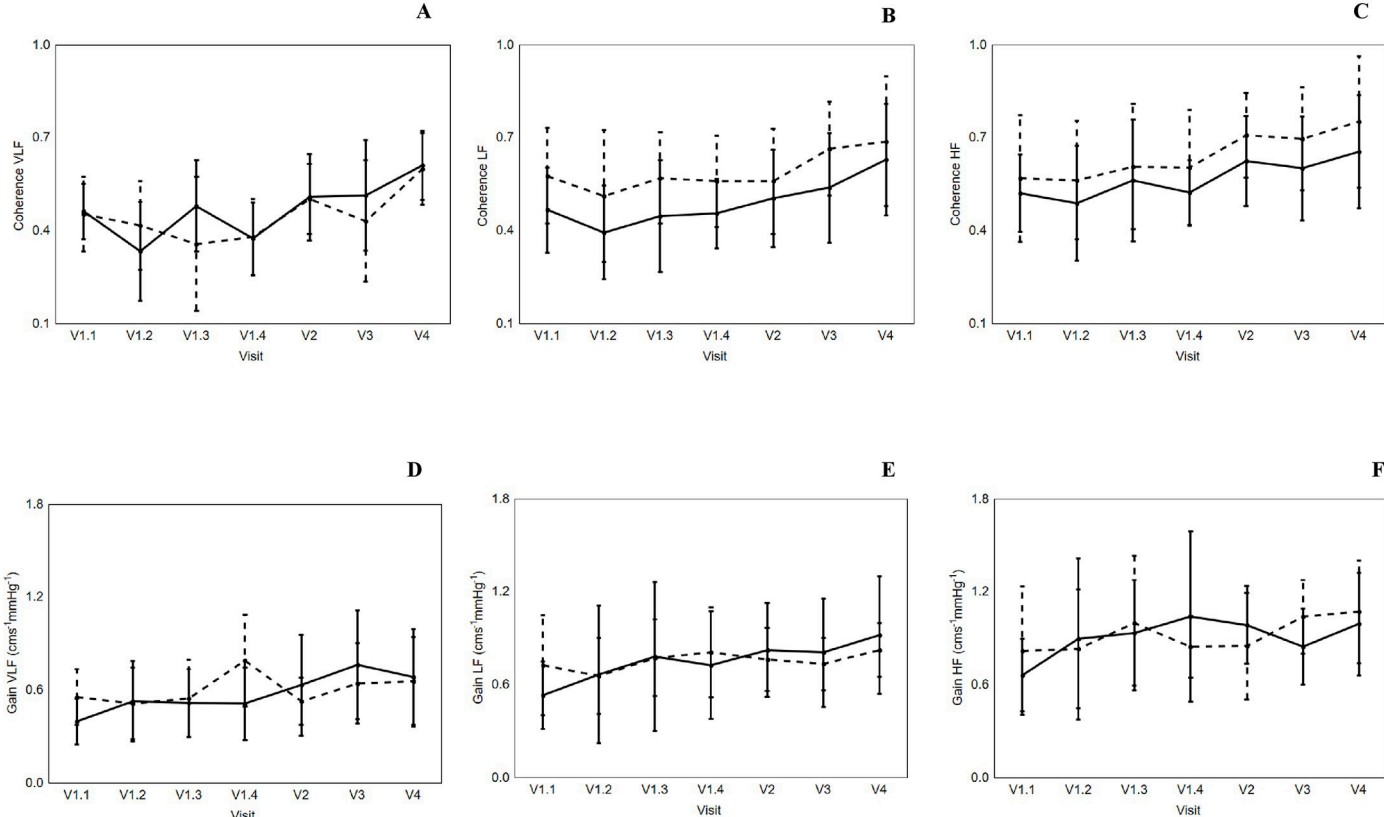

**Fig 2.** Effects of visit on the changes in Coherence (A, B and C) and Gain (D, E and F). Very low frequency (VLF, A and D) versus low frequency (LF, B and E) versus high frequency (HF, C and F). Affected hemisphere (continuous line) and non-affected hemisphere (dotted line). Vertical bar denotes 95% confidence interval. V1.1 and V1.2 refer to during IVT. V1.3 refers to immediately prior to end of IVT, and Visit 1.4 refers to immediately after the end of IVT. V2, V3 and V4 refers to 24 hours, 2 weeks and 3 months post stroke symptom onset, respectively.

changes in cerebrovascular resistance; both parameters reflecting changes in the instantaneous CBV-BP relationship. Previous studies reported that CrCP was inversely associated with $PaCO_2$ [22, 33, 44, 45]. Grune and colleagues carried out a study on 10 patients who underwent general anesthesia and reported that hypocapnia led to an increment in CrCP [45]. In this study, the concurrent increment of $ETCO_2$ and further exacerbation by BP reduction observed over the course of the disease, suggests that these parameters are likely to be interrelated. On the other hand, we did not demonstrate any significant changes in AH or NAH RAP either during or immediately after the completion of IVT, nor were there significant changes in AH or NAH RAP over subsequent follow-up visits. Unlike CrCP, the relationship between RAP and $ETCO_2$ is not clear. Some studies have not found an association between Rap and $ETCO_2$ [22, 46]. Future studies should consider investigating the regulatory mechanism between these factors.

Though it is known that dCA is effective in the VLF and LF regions, we have also reported values in the HF band as recommended by the CARNet white paper [23]. No differences in terms of temporal pattern were observed in coherence and gain in both AH and NAH at different frequency bands and visits. The reduction of LF phase observed in AH, during the early phase of IVT, is an important finding, indicating worsening of dCA at this crucial stage of treatment. Subsequent alterations in dCA, following IVT, might have been obscured by the relatively wide scattering of phase shift values, reflecting limitations in statistical power, as discussed below.

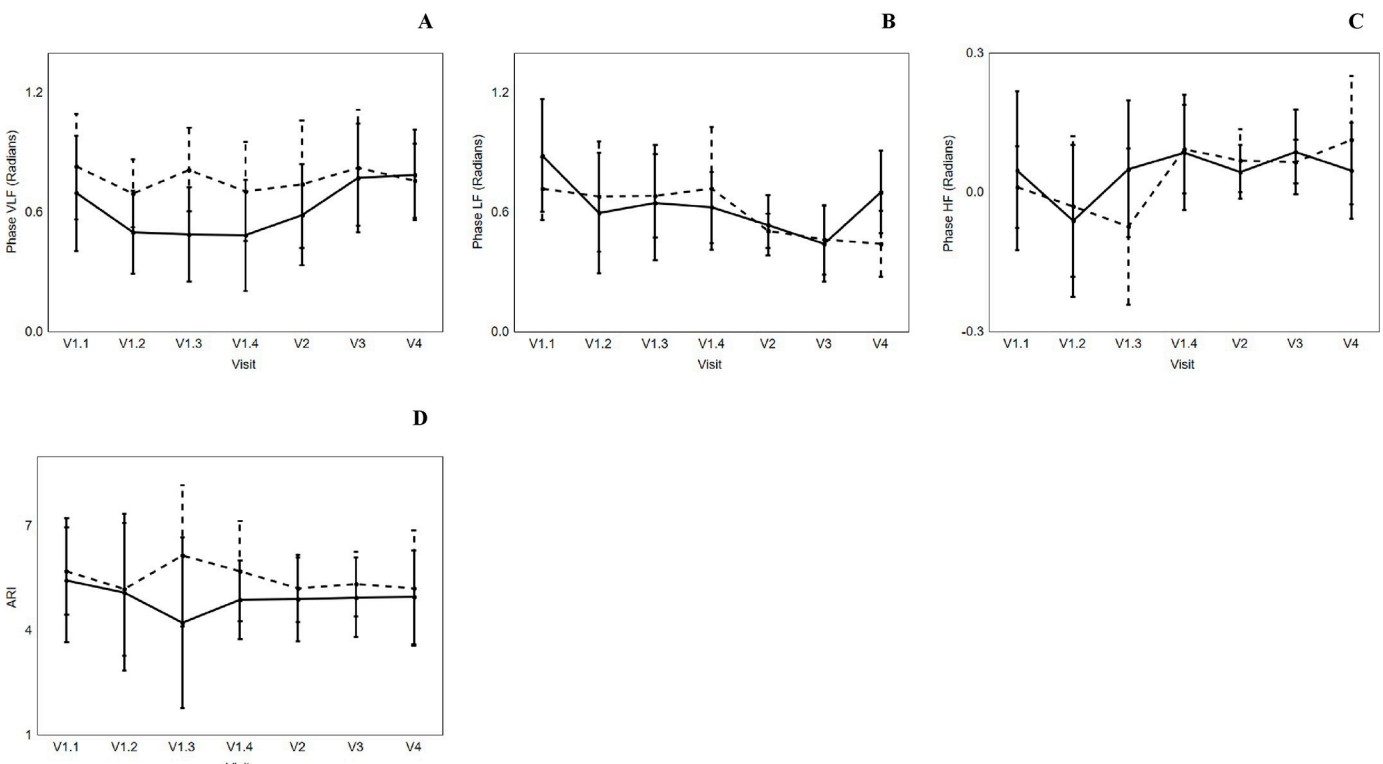

**Fig 3.** Effects of visit on the changes in Phase (A, B and C) and autoregulation index (ARI) (D). Very low frequency (VLF, A); low frequency (LF, B); and high frequency (HF, C). Affected hemisphere (continuous line) and non-affected hemisphere (dotted line). Vertical bar denotes 95% confidence interval. V1.1 and V1.2 refer to during IVT. V1.3 refers to immediately prior to end of IVT, and Visit 1.4 refers to immediately after the end of IVT. V2, V3 and V4 refers to 24 hours, 2 weeks and 3 months post stroke symptom onset, respectively.

Although the ARI dropped from 5.45 ± 2.65 (Visit 1.1) to 4.23 ± 2.65 (Visit 1.3), we did not test the significance of this difference as the F-test for the intra-visit ANOVA was non-significant (Table 6, p = 0.17). Given the significant reduction in phase observed for the same time intervals (Table 5), we can speculate that the ARI did not show a statistically similar result due to the inconsistent effects of gain (Table 4), since both phase and gain contribute to the estimation of the ARI. Moreover, there were only a small number of AIS patients (n = 3) with impaired dCA in this study, as indicated by ARI<4 [47] at Visit 1.1. Fig 4A and 4B display the ARI values of AIS patients across all visits and it can be seen that such participants behave

**Table 6. ARI in AIS patients receiving IVT.**

| Parameters | Visit 1.1 | Visit 1.2 | Visit 1.3 | Visit 1.4 | Visit 2 | Visit 3 | Visit 4 | P value (Intra-visit effect, Visit 1.1–1.4) | P value (Inter-visit effect, Visit 1.4–4) | P value (Hemispheric effect, Visit 1.1–1.4) | P value (Hemispheric effect, Visit 1.4–4) |
|---|---|---|---|---|---|---|---|---|---|---|---|
| **ARI, AH** | 5.45 ± 2.65 | 5.10 ± 3.15 | 4.23 ± 2.65 | 4.88 ± 1.69 | 4.91 ± 1.79 | 4.95 ± 1.36 | 4.96 ± 1.46 | 0.99 | 0.89 | 0.17 | 0.68 |
| **ARI, NAH** | 5.71 ± 1.64 | 5.19 ± 2.48 | 6.16 ± 1.63 | 5.72 ± 1.87 | 5.22 ± 1.26 | 5.34 ± 1.01 | 5.23 ± 1.79 | 0.59 | 0.30 | | |

Data are mean ± SD. P-values from two-way ANOVA for effects of during and immediately post-IVT (Intra-visit: Visit 1.1 to Visit 1.4) and subsequent visit (Inter-visit: Visit 1.4 to Visit 4) (Visit effect) and for the effects of hemisphere (AH vs. NAH). AH, affected hemisphere; AIS, acute ischemic stroke; ARI, autoregulation index; IVT, intravenous thrombolysis; NAH, non-affected hemisphere. Visit 1.1 and Visit 1.2 refer to during IVT, Visit 1.3 refers to immediately prior to end of IVT, and Visit 1.4 refers to immediately after the end of IVT. Visit 2, Visit 3, and Visit 4 refer to approximately 24 hours, 2 weeks, and 3 months post stroke symptom onset, respectively.

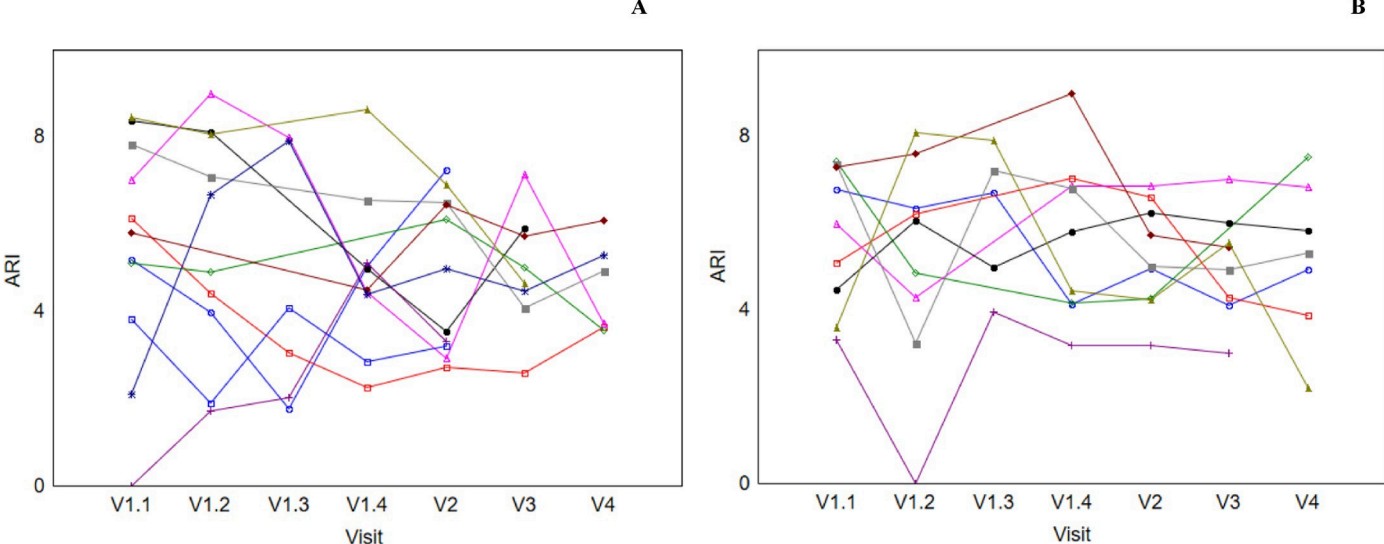

**Fig 4. Individual subjects' autoregulation index (ARI) values at each visit.** [A] AH ARI vs. [B] NAH ARI. V1.1 and V1.2 refer to during IVT. V1.3 refers to immediately prior to end of IVT, and V1.4 refers to after the end of IVT. V2, V3 and V4 refers to approximately 24 hours, 2 weeks and 3 months post stroke symptom onset, respectively. ARI, autoregulation index; AH, affected hemisphere; IVT, intravenous thrombolysis; NAH, non-affected hemisphere.

differently during the course of the disease. As the sample size of the study cohort is relatively small, it is difficult to draw any meaningful conclusions, especially as the study was designed to detect a change of ARI of 2 units. Of note, a substantial increase in sample size to n = 44 would be needed to be able to detect changes in ARI of 1 unit [26]. Further investigations should consider recruiting a larger number of patients, with higher NIHSS scores in order to assess any relationships between changes of such parameters in AIS patients, in particular to those who present secondary reperfusion injury or poorer functional outcome.

## Limitations

Our study has a number of limitations. Firstly, due to the study design and to avoid any delays in IVT administration, we could not perform TCD measurement on AIS patients prior to thrombolysis treatment. Therefore, we lost critical information regarding cerebral hemodynamic status prior to IVT therapy, though initial assessments were performed within 20 minutes of commencing IVT. Secondly, as a pilot study, angiographic imaging was not available. Assumption was made that successful reperfusion would allow presence of the MCA TCD signal, and non-recanalized participants, who are important and account for a significant proportion of AIS patients who receive IVT, were therefore not able to be included in the study. It is possible that non M1 MCA occlusion stroke, including LACS or POCS, would have detectable MCA signals regardless of the recanalization status. Nonetheless, improved NIHSS and mRS scores in our patient cohort would indicate that satisfactory reperfusion was achieved. Future studies should consider using alternative imaging modalities (e.g. CT angiography) in order to investigate this important, and yet often overlooked cohort. Thirdly, we had a relatively small sample size with significant heterogeneity in stroke subtype. As a result, it is difficult to perform sub-group analysis and the positive and negative results we observed in this study could be due to Type I and II errors, respectively. Furthermore, an admission mean NIHSS of 8.2, indicated that the majority of our AIS patients tended to have mild/ moderate stroke disease. Moreover, a median mRS of 2 indicated that the majority of the participants had a benign disease course with a satisfactory recovery at 3 months post symptom onset. Therefore, it would

be beneficial for future studies to recruit a larger cohort, with higher NIHSS scores, in order to provide a better insight on the time course and changes in CA in AIS patients deemed suitable to receive IVT and their associated functional outcome. Fourthly, TCD was used to measure CBV, as a surrogate of CBF, under the assumption that the insonated vessel diameter remains constant [48]. Since TCD can only reflect CBV in the MCA or another large intra-cranial vessel, it can only provide global hemispheric values. Therefore, should CA impairment occur in a more localised, smaller infarct area or collaterals in the penumbra, TCD may be insensitive to provide information of such focal changes. Finally, with increasing use of MT with and without co-commitment IVT in AIS patients, it would be useful to carry out similar studies, to investigate how CA changes pre- and post-MT in AIS.

## Conclusion

The feasibility of using TCD to assess dynamic CA in AIS patients during IVT, and up to 3 months following stroke was demonstrated in this pilot study. A reduction in LF phase values in AH suggests that impairment of dCA could occur during IVT, and may warrant closer monitoring at such a critical stage. Functional CA status could be considered as part of routine clinical care, as it may provide valuable additional information about guidance of antihypertensive treatment and response to IVT or other interventions. However, the prolonged time course of CA evolution changes in AIS warrant further investigation, with a larger cohort and a wider range of stroke severity, to allow better patient risk stratification and assessment of outcome following IVT.

## Supporting information

**S1 Data.**
(XLSX)

## Acknowledgments

The authors thank all study participants for their time and effort they devoted to this study.

## Author Contributions

**Conceptualization:** Man Y. Lam, Victoria J. Haunton, Ronney B. Panerai, Thompson G. Robinson.

**Data curation:** Man Y. Lam.

**Formal analysis:** Man Y. Lam.

**Investigation:** Man Y. Lam.

**Methodology:** Man Y. Lam.

**Supervision:** Victoria J. Haunton, Ronney B. Panerai, Thompson G. Robinson.

**Writing – original draft:** Man Y. Lam.

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
