## [Decision Letter · Decision Letter 0]

1 Jul 2020

PONE-D-20-16644

Cerebral Hemodynamics in Stroke Thrombolysis (CHiST) Study

PLOS ONE

Dear Dr. Man Yee Lam

Thank you for submitting your manuscript to PLOS ONE. After careful consideration, we feel that it has merit but does not fully meet PLOS ONE’s publication criteria as it currently stands. Therefore, we invite you to submit a revised version of the manuscript that addresses the points raised during the review process.

I would appreciate if pay careful attention to the reviewer's comments in your revised manuscript.

We look forward to receiving your revised manuscript.

Kind regards,

Ehab Farag, MD FRCA FASA

Academic Editor

PLOS ONE

Journal Requirements:

'Professor Thompson G Robinson is the NIHR senior investigator, otherwise, no other

authors have competing interests.'.

Please include your updated Competing Interests statement in your cover letter; we will change the online submission form on your behalf

4. Please include a copy of Tables 1 to 6 which you refer to in your text.

Additional Editor Comments (if provided):

Reviewers' comments:

Reviewer's Responses to Questions

**Comments to the Author**

1. Is the manuscript technically sound, and do the data support the conclusions?

Reviewer #1: Yes

2. Has the statistical analysis been performed appropriately and rigorously? 

Reviewer #1: Yes

3. Have the authors made all data underlying the findings in their manuscript fully available?

Reviewer #1: Yes

4. Is the manuscript presented in an intelligible fashion and written in standard English?

Reviewer #1: Yes

5. Review Comments to the Author

Reviewer #1: 1. What are the main claims of the paper and how significant are they for the discipline?

Main purpose of the paper was to conduct a pilot study in intravenous thrombolysis patients to detect changes in cerebral autoregulation during and following successful recanalization at 4 time points in the acute, subacute and chronic stages after an acute ischemic stroke. Transcranial doppler data sets, blood pressure, and endtidal Co2 were monitored.

This pilot study is significant for the discipline given a knowledge gap in understanding the patterns of cerebral autoregulation in various stroke patterns at different time points including the chronic phase and the role the data plays in reperfusion injury after recanalization.

2. Are the claims properly placed in the context of the previous literature? Have the authors treated the literature fairly?

Literature was treated fairly.

3. Do the data and analyses fully support the claims? If not, what other evidence is required?

Data analyses revealed a reduction in BP during intravenous thrombolysis. Reduction in heart rate, reduction in critical closing pressure in the affected and non-affected hemisphere were observed post iv thrombolysis across visits to the 3-month mark. Endtidal CO2 increased during subacute and chronic stages. No changes in cerebral blood velocity, coherence, gain, and Autoregulation Index during the follow-up period.

It would be important to mention whether any or no medication management was applied in the acute phase and in the post-stroke chronic phase while the patients were evaluated. E.g. change in antihypertensive medication, new use of antiarrhythmic drugs (beta blockade for arrhythmia or tachycardia), application or need for supplemental oxygen or anxiolytic, analgesic medication with or without supplemental oxygen, especially when the patient’s airway maintenance was at risk or the patient was agitated. Patients’ altered spontaneous breathing efforts can interfere with systemic blood pressure and cerebral perfusion pressure, as well as fluctuation of etCO2 .

4. PLOS ONE encourages authors to publish detailed protocols and algorithms as supporting information online. Do any particular methods used in the manuscript warrant such treatment? If a protocol is already provided, for example for a randomized controlled trial, are there any important deviations from it? If so, have the authors explained adequately why the deviations occurred?

Authors did follow a detailed protocol, all methods were addressed, no deviations apparent for reviewer. 2 patients were excluded with difficulties to obtain the appropriate TCD windows. This reduced the patient group to 11 study patients and limited the ability to address subgroups.

5. If the paper is considered unsuitable for publication in its present form, does the study itself show sufficient potential that the authors should be encouraged to resubmit a revised version?

I recommend considering this pilot study for publication. The discussion includes reasons why more data is required after this pilot study to better understand the effects of changes in cerebral blood pressure/systemic blood pressure affecting reperfusion injury.

6. Are original data deposited in appropriate repositories and accession/version numbers provided for genes, proteins, mutants, diseases, etc.?

Yes

7. Are details of the methodology sufficient to allow the experiments to be reproduced?

Yes

8. Is the manuscript well organized and written clearly enough to be accessible to non-specialists?

Yes. I do compliment the research team for their candid data review and laying out the limitations of this small pilot study and their interest to address these limitations in a larger cohort to optimize individualized patient care. The manuscript is written clearly, though methods and techniques applied use specific terminology that the non-specialist may not be familiar with, however use of proper terminology cannot be avoided in this context.

6. PLOS authors have the option to publish the peer review history of their article (what does this mean?). If published, this will include your full peer review and any attached files.

Reviewer #1: **Yes: **Dorothea Rosenberger MD PhD

---

## [Author Response · Author response to Decision Letter 0]

14 Aug 2020

Journal Requirements:

Response: Many thanks for these instructions. We confirmed we have read the PLOS ONE’s style requirements carefully and uploaded the revised manuscript, cover letter and figures to the website accordingly. 

Response: Thank you for the guidance. We confirm there are no restrictions to data access and we have uploaded the minimal anonymized data set as Supporting Information files.

'Professor Thompson G Robinson is the NIHR senior investigator, otherwise, no other

authors have competing interests.'.

Response: We have confirmed that ‘Professor Thompson G Robinson is the NIHR senior investigator, otherwise, no other authors have competing interests, this does not alter our adherence to PLOS ONE policies on sharing data and materials’ on revised manuscript (Tracked version: page 1; Clean version: page 1). 

Response: We confirm we have updated the competing interest statement in our revised cover letter. 

4. Please include a copy of Tables 1 to 6 which you refer to in your text.

Response: Many thanks for the comment, we have included a copy of Tables 1 to 6 in the revised manuscript.

Response: This can be confirmed as well. We have included our raw data on the Supporting Information Files. 

Additional Editor Comments (if provided):

Reviewers' comments:

Reviewer's Responses to Questions

Comments to the Author

1. Is the manuscript technically sound, and do the data support the conclusions?

Reviewer #1: Yes

2. Has the statistical analysis been performed appropriately and rigorously? 

Reviewer #1: Yes

3. Have the authors made all data underlying the findings in their manuscript fully available?

Reviewer #1: Yes

4. Is the manuscript presented in an intelligible fashion and written in standard English?

Reviewer #1: Yes

5. Review Comments to the Author

Reviewer #1: 1. What are the main claims of the paper and how significant are they for the discipline?

Main purpose of the paper was to conduct a pilot study in intravenous thrombolysis patients to detect changes in cerebral autoregulation during and following successful recanalization at 4 time points in the acute, subacute and chronic stages after an acute ischemic stroke. Transcranial doppler data sets, blood pressure, and endtidal Co2 were monitored.

This pilot study is significant for the discipline given a knowledge gap in understanding the patterns of cerebral autoregulation in various stroke patterns at different time points including the chronic phase and the role the data plays in reperfusion injury after recanalization.

2. Are the claims properly placed in the context of the previous literature? Have the authors treated the literature fairly?

Literature was treated fairly.

3. Do the data and analyses fully support the claims? If not, what other evidence is required?

Data analyses revealed a reduction in BP during intravenous thrombolysis. Reduction in heart rate, reduction in critical closing pressure in the affected and non-affected hemisphere were observed post iv thrombolysis across visits to the 3-month mark. Endtidal CO2 increased during subacute and chronic stages. No changes in cerebral blood velocity, coherence, gain, and Autoregulation Index during the follow-up period.

It would be important to mention whether any or no medication management was applied in the acute phase and in the post-stroke chronic phase while the patients were evaluated. E.g. change in antihypertensive medication, new use of antiarrhythmic drugs (beta blockade for arrhythmia or tachycardia), application or need for supplemental oxygen or anxiolytic, analgesic medication with or without supplemental oxygen, especially when the patient’s airway maintenance was at risk or the patient was agitated. Patients’ altered spontaneous breathing efforts can interfere with systemic blood pressure and cerebral perfusion pressure, as well as fluctuation of etCO2 .

Response: Many thanks for this important comment. We have confirmed that none of the patients required antihypertensive medication or analgesia during IVT infusion. One patient required supplemental oxygen (1 litre via nasal cannula) during IVT infusion. 4 patients were discharged with additional antihypertensive medication, with two additional patients receiving antihypertensive medication at visit 4 (3 months post stroke symptom onset) (Tracked version: page 10; Clean version: page 9). 

4. PLOS ONE encourages authors to publish detailed protocols and algorithms as supporting information online. Do any particular methods used in the manuscript warrant such treatment? If a protocol is already provided, for example for a randomized controlled trial, are there any important deviations from it? If so, have the authors explained adequately why the deviations occurred?

Authors did follow a detailed protocol, all methods were addressed, no deviations apparent for reviewer. 2 patients were excluded with difficulties to obtain the appropriate TCD windows. This reduced the patient group to 11 study patients and limited the ability to address subgroups.

5. If the paper is considered unsuitable for publication in its present form, does the study itself show sufficient potential that the authors should be encouraged to resubmit a revised version?

I recommend considering this pilot study for publication. The discussion includes reasons why more data is required after this pilot study to better understand the effects of changes in cerebral blood pressure/systemic blood pressure affecting reperfusion injury.

6. Are original data deposited in appropriate repositories and accession/version numbers provided for genes, proteins, mutants, diseases, etc.?

Yes

7. Are details of the methodology sufficient to allow the experiments to be reproduced?

Yes

8. Is the manuscript well organized and written clearly enough to be accessible to non-specialists?

Yes. I do compliment the research team for their candid data review and laying out the limitations of this small pilot study and their interest to address these limitations in a larger cohort to optimize individualized patient care. The manuscript is written clearly, though methods and techniques applied use specific terminology that the non-specialist may not be familiar with, however use of proper terminology cannot be avoided in this context.

6. PLOS authors have the option to publish the peer review history of their article (what does this mean?). If published, this will include your full peer review and any attached files.

Do you want your identity to be public for this peer review? For information about this choice, including consent withdrawal, please see our Privacy Policy.

Reviewer #1: Yes: Dorothea Rosenberger MD PhD

---

## [Editor Report · Decision Letter 1]

21 Aug 2020

Cerebral Hemodynamics in Stroke Thrombolysis (CHiST) Study

PONE-D-20-16644R1

Dear Dr.

Man Yee Lam

We’re pleased to inform you that your manuscript has been judged scientifically suitable for publication and will be formally accepted for publication once it meets all outstanding technical requirements.

Kind regards,

Ehab Farag, MD FRCA FASA

Academic Editor

PLOS ONE
---

## [Editor Report · Acceptance letter]

27 Aug 2020

PONE-D-20-16644R1 

Cerebral hemodynamics in stroke thrombolysis (CHiST) study 

Dear Dr. Lam:

I'm pleased to inform you that your manuscript has been deemed suitable for publication in PLOS ONE. Congratulations! Your manuscript is now with our production department. 

Kind regards, 

on behalf of

Dr. Ehab Farag 

Academic Editor

PLOS ONE